# Quantifying wedge-tailed shearwater (*Ardenna pacifica*) fallout after changes in highway lighting on Southeast Oʻahu, Hawaiʻi

**Jennifer Urmston** [1,2¤]*, **K. David Hyrenbach** [1,2], **Keith Swindle** [3]

**1** Hawaiʻi Pacific University, Waimānalo, HI, United States of America, **2** Oikonos Ecosystem Knowledge, Kailua, HI, United States of America, **3** U.S. Fish and Wildlife Service (USFWS), U.S. Embassy, Nairobi, Kenya

¤ Current address: Migratory Bird Permit Office, U.S. Fish and Wildlife Service (USFWS), Portland, OR, United States of America

* Jennifer_Urmston@fws.gov

**Data Availability Statement:** All relevant data are within the manuscript and its Supporting Information files.

## Abstract

Attraction to artificial light at night (ALAN) poses a threat to many fledgling seabirds leaving their nests for the first time. In Hawaiʻi, fledgling wedge-tailed shearwaters disoriented by lights may become grounded due to exhaustion or collision, exposing them to additional threats from road traffic and predation. While the timing and magnitude of shearwater fallout varies from year to year, little is known about how changing lighting and environmental conditions influence the risk of grounding for this species. We analyzed 8 years (2012–2019) of observations of road-killed shearwaters along the Kalanianaʻole Highway on Oʻahu to quantify the timing and magnitude of fallout during the fledging season (November–December). Our goal was to compare fallout before (2012–15) and after (2016–19) a transition in highway lighting from unshielded high-pressure sodium (HPS) to full-cutoff light-emitting diode (LED) streetlights. To detect the shearwater response to the lighting regime, we also accounted for three potential environmental drivers of interannual variability in fallout: moon illumination, wind speed, and wind direction. The effects of these environmental drivers varied across years, with moon illumination, wind speed and wind direction significantly affecting fallout in at least one year. Altogether, the interaction between moon illumination and wind speed was the most important predictor, suggesting that fallout increases during nights with low moon and strong winds. The lack of an increase in fallout after the change from HPS to shielded 3000K - 4000K LED streetlights suggests the new streetlights did not worsen the light pollution impacts on wedge-tailed shearwaters on Southeast Oʻahu. However, due to potential species-specific disparities in the behavior and light attraction of petrels, similar studies are needed before energy saving LED lights are implemented throughout the Hawaiian archipelago.

## Introduction

Light pollution is a concern for burrow-nesting seabirds globally, with documented impacts on over 50 species of shearwaters, petrels, and puffins [1]. While coastal light pollution can

**Funding:** This work was supported by Experiment.
com (Blinded by the light: reducing shearwater
deaths along a coastal highway in O 'ahu, Hawai'i)
and The Eppley Foundation for Research (Blinded
by the Light: Shearwater Deaths Along a Coastal
Highway in O'ahu). The funders had no role in
study design, data collection and analysis, decision
to publish, or preparation of the manuscript.

**Competing interests:** The authors have declared
that no competing interests exist.

disrupt adult seabirds provisioning their chicks on colonies [2–4], fledglings consistently account for the majority (68% - 99%) of the grounded specimens [1]. Fledgling seabird "fallout" occurs when chicks leaving their nests are disoriented by onshore lighting and become stranded on land instead of flying out to sea [1]. The magnitude of fallout is likely influenced by the number of chicks fledging, the prevailing environmental and celestial conditions [4, 5], and the features of anthropogenic lights, which vary as a function of light fixture design and bulb type [6, 7]. To gauge the effectiveness of light pollution mitigation measures, wildlife managers need to understand the influence of these biological and environmental drivers on the timing and magnitude of fallout.

A conceptual model to explain fallout involves fledging seabirds being drawn toward well-lit coastal areas, especially in the absence of moonlight [4, 5, 8–11] and when strong winds are directed toward shore [10, 11]. Birds are affected by bright light sources from vessels at sea and urbanized areas on shore, including streetlights and sports fields [1, 6, 12–13]. Moreover, collisions with powerlines and other structure can lead to injury and grounding [14, 15]. While our understanding of the environmental drivers of fallout is growing, the influence of specific design features of anthropogenic light sources remains understudied. In particular, lamp color and directionality are two key streetlight features that can affect fallout [7, 16]. Spurred by efforts to improve energetic efficiency, many cities are replacing yellow high-pressure sodium (HPS) lightbulbs commonly used in streetlights with white light-emitting diode (LED) bulbs [17–19]. Although LED bulbs decrease electricity consumption and maintenance costs, these benefits could be costly to wildlife, as shearwaters may be more sensitive to LED lights [7]. A study on the visual perception of Wedge-tailed Shearwaters (*Ardenna pacifica*–previously *Puffinus pacificus)* showed that they experience maximum light absorption of the wavelengths emitted by white LED lights (406–566 nm) and have lower absorption of the wavelengths emitted by HPS lights (560–620 nm) [20]. Moreover, a field-based experiment in Australia showed that Short-tailed Shearwaters (*Ardenna tenuirostris*) show increased attraction to LED lights over HPS lights, although the difference was not statistically significant [7].

Mitigation measures often target light directionality, whereby streetlights are shielded through the use of a "full-cutoff" design, which inhibits light emission above the horizontal plane of the fixture. This approach, when applied to HPS lights, reduced Newell's Shearwater (*Puffinus newelli*) fallout on Kauai (Hawai'i) [16]. Although mitigation is being addressed through shielding, the common use of optimized LEDs with broad spectra and Correlated Color Temperature (CCT) greater than the maximum recommended value for wildlife (2200 K) may be a cause for concern [17]. While modern LED lights possess the flexibility to give off a range of low to high CTT, short-wavelength light with high CCT is a common choice because of its efficiency [19]. The effectiveness of light shielding coupled with the use of broad spectrum, high CCT LEDs is unknown.

On the island of Oʻahu (Hawaiʻi), Wedge-tailed Shearwaters (hereafter referred to as WTSH) experience fallout during the annual fledging season (November-December) [21, 22]. A three-year study in the early 1990s, revealed that hundreds of chicks become grounded every autumn, with the number varying widely from year to year [22]. Starting in 2002, U.S. Fish and Wildlife Service initiated a program of opportunistic road surveys of the southeast section of Oʻahu during the fledging season, which documented a fallout hotspot in the town of Waimānalo, within 5 km from two WTSH colonies on offshore islets [21].

While there is evidence of interannual variability in WTSH fallout, little is known about the influence of environmental (weather and oceanographic conditions) and biological (breeding population size and reproductive success) drivers. To date, only one study has investigated the environmental drivers of WTSH fallout, by comparing a "wreck" year of unusually high fallout (1994), when WTSH groundings increased ten-fold from the two "normal" years prior [22].

This study suggested that anomalous southerly winds likely carried fledglings inland rather than out to sea and scattered them throughout the windward coast of Oʻahu. While the southerly winds help explain why many birds were found inland, it is unclear to what extent low ocean productivity during the breeding season and unusual weather conditions during the fledging period caused the high fallout observed that year.

Over a decade later, Friswold et al. (2020) documented an increasing trend in annual fallout numbers between 2003 and 2010, and a two-year cycle of alternating years of high and low fallout. Subsequently, an unusually large fallout event in 2011 was documented during a La Niña year with high ocean productivity [23]. These results are suggestive of the potential influence of breeding population size and reproductive success on fallout.

In 2012, we began conducting systematic road surveys along a 17.3-km section of the Kalanianaʻole Highway to document WTSH fallout. In 2016, the Hawaiʻi Department of Transportation changed the streetlights on Oʻahu's major roads from unshielded 2200 K HPS lights to shielded 3000–4000 K LED lights, where Kelvin (K) is a unit of measurement for CCT. Lower CCT indicates a warm yellow-orange appearance whereas higher CCT indicates cool blue light [18]. The shift in lights halfway through our study provided a unique opportunity to compare WTSH fallout under different street lighting conditions. To this end, we continued conducting standardized surveys following the established protocol through 2019 and analyzed an 8-year time series with four years before (2012–15) and four years after (2016–19) the change in lighting. This is the first study to compare changes in seabird groundings in response to HPS versus LED streetlights, by repeatedly surveying a fallout hotspot during the fledging season.

The goal of this study is to quantify the magnitude of WTSH fallout under two contrasting lighting regimes, to inform future coastal development and management of light pollution. Although shielding of the LED streetlights may reduce initial WTSH attraction, we predicted that disorientation caused by high intensity/shorter wavelength lights would outweigh the benefits of shielding. Thus, we expected an increase in fallout after the installation of LED streetlights (2016–2019). To detect the fallout response to the lighting regime, we also accounted for three potential environmental drivers: moon illumination, wind speed, and wind direction. Because WTSH rely on wind to take flight and may become disoriented in the absence of moonlight, we predicted higher fallout during windy nights of low moon illumination. In particular, due to the location of our study area, southwest from two breeding colonies, we anticipated that strong northeasterly winds would drive the fledging birds towards shore.

## Methods

### Study area

This study focuses on the southeast section of Oʻahu, where a two-lane coastal highway runs through a rural and developed landscape (Fig 1). The survey route was illuminated with HPS streetlights until 2016, when the Hawaiʻi Department of Transportation transitioned to LED streetlights. The CCT of the LED streetlights is 3000 K on sections of the highway directly adjacent to the ocean, whereas inland lights are 4000 K.

The WTSH breeding colonies of Mānana Island and Kāohikaipu Island, where approximately 25,000 and 800 chicks were counted in 2019, are located 1.3 and 0.7 km offshore of our study area, respectively [25]. Three additional WTSH colonies on offshore islets (Mokulua Nui, Mokulua Iki, and Popoiʻa) lie approximately 6 km north of the study area (Fig 1), with 2019 chick count estimates of 3,500, 5,000, and 900, respectively [25].

Weather patterns on windward Oʻahu are dominated by the northeast trade winds, which typically persist for 1 to 2 weeks, interspersed with no-wind periods or southerly storms. Peak wind speeds occur in the afternoon, with lower wind speeds at night [26].

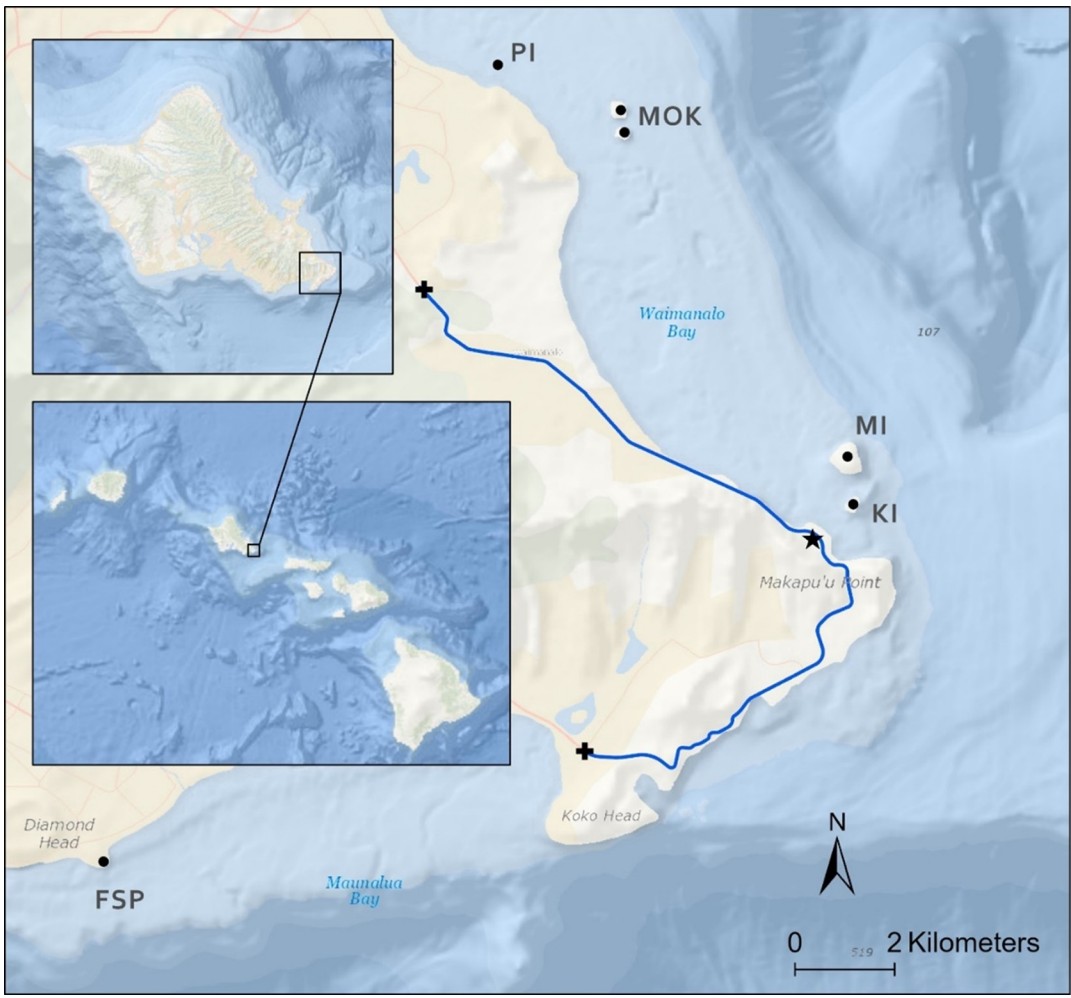

**Fig 1. Map of the study area in southeast Oʻahu.** Blue line shows the survey route, and crosses indicate the start and end points. Black dots indicate WTSH breeding colonies. (PI = Popoiʻa Island, MOK = Mokulua Islands, MI = Mānana Island, KI = Kāohikaipu Island, FSP = Freeman Seabird Preserve). Star marks the location of the Sea Life Park seabird rehabilitation center. Inset maps show the island of Oʻahu, and the main Hawaiian Islands. Map features are overlaid on an ArcGIS Pro Software Version 2.5 base layer [24].

### Intake records

Members of the public deliver grounded WTSH to Sea Life Park (SLP), a marine life center located along our survey route in Waimānalo, for rescue and rehabilitation. SLP intake records, involving the daily number of rescued WTSH chicks, have been used to document the island-wide temporal variability in WTSH fallout during the fledging season and from year to year [21, 22]. To provide a broader context for our localized surveys of a known WTSH fallout hotspot, we compared the timing and the magnitude of annual fallout documented in the SLP intake records and our surveys.

### Road surveys

We used a time series of standardized road surveys along a 17.3-km stretch of the Kalania-naʻole Highway, starting at the Olomana Golf Club, running through Waimānalo, and ending at the Koko Marina Center (Fig 1). While this survey route is a subset of the area surveyed by

USFWS from 2002–2010, it encompasses the main WTSH fallout hotspot in Waimānalo [21]. We conducted morning surveys by car, every 3 days, throughout the WTSH fledging season (November 6 –December 21). We began surveys at sunrise (6:15–7:15 AM) and drove the route once in each direction, at speeds between 25–35 mph, while visually searching for dead birds in each lane and along the shoulder. Since these surveys were conducted in the morning, likely a full 12 hours after fledging time, almost all the birds we observed were deceased. In 8 years of surveys, we observed 2 live birds, which were brought to SLP for rehabilitation and not counted in our analysis. All dead birds sighted while driving were included in the surveys, even if they were found on the shoulder, the median, or off the road.

Upon encountering a carcass, we recorded its position on the road, location (latitude and longitude coordinates from a hand-held Garmin e-trex GPS unit), nearest street address, and nearest utility pole using their unique id tags. We also took photos of each WTSH we encountered, showing diagnostic identification features (head and feet).

## Environmental variables

We related WTSH fallout to two publicly-available environmental datasets: (i) wind speed (knots) and wind direction (degrees) recorded on Moku Loʻe (Kaneohe Bay) and provided by PacIOOS [27], and (ii) the lunar cycle, quantified using the percent of the lunar disk that was illuminated each night, from the U.S. Naval Observatory [28].

Because WTSH fledge during the night, we averaged the hourly wind data every night (18:00–6:00 local time). To match our surveys to the preceding environmental conditions, we related the number of grounded WTSH documented during a given road survey to the average wind speed (knots), wind direction (degrees), and lunar disk illumination (%) from the three nights prior.

## Data analysis

We analyzed fallout across and within years using generalized linear models (GLM) built with R version 3.5.1 and the stats and MASS packages [29]. We developed and fitted nine separate models: a full model (involving all study years) quantified interannual variability, and eight yearly models visualized the interannual differences documented by the full model.

We ran all models using both Poisson and negative binomial distributions. Because the Poisson assumes that the variance equals the mean, the negative binomial is more appropriate whenever there is overdispersion [30]. We used the Akaike Information Criterion corrected for small sample size (AICc) to select the best-fitting distribution for each model [31].

**Multi-year model of WTSH fallout.** We related the number of WTSH observed during 128 surveys of the entire study area (16 per year times 8 years) to the light regime (unshielded HPS / shielded LED), year (2012–2019), and four environmental variables: moon illumination (% lunar disk illuminated), average wind speed (knots), average wind direction (degrees), and Julian date (the number of days since the beginning of the year).

We used multi-model inference to test all possible combinations of these six explanatory variables. Whenever two of three potentially interacting variables were included in a model, we also considered their interaction ('moon*date', 'moon*wind speed', and 'wind speed*date'). We used the AICmodavg package [29] to assess the model fit using AICc, which prevents over-fitting by penalizing models for each additional variable [31]. AICc assigns a value to each model using the formula, AICc = $-2log(L)+2K+(2K(K+1)/(n\text{-}K\text{-}1))$, where $K$ is the number of parameters, $n$ is the sample size, and $L$ is the maximum likelihood of obtaining the given results with $K$ parameters. We used Akaike weights ($w_i$) to calculate the likelihood of each

model as follows:

$$\omega_i = \frac{exp\left(\frac{-\Delta_i}{2}\right)}{\Sigma_{r=1}^{R} exp\left(\frac{-\Delta_r}{2}\right)}$$

where the numerator is the model likelihood with $\Delta_i$ showing the change from the lowest AICc model to the given model, and the denominator is the sum of all relative weights, as determined by $\Delta_r$, the change in each contending model from the lowest AICc model. The lowest AICc value indicates the model that best describes the patterns in the observed data without over-fitting [31, 32].

To test the influence of the streetlights, in the context of interannual variability, we built two complementary sets of full models that either included "light regime" (comparing two groups of years: 2012–15 vs 2016–19) or included individual "years", regardless of their "light regime". This resulted in a total of 106 models: 36 included "light regimes", 36 included "years", and 34 included neither. Individual models ranged from having one to eight predictors (five variables and three interactions) (S1 Table). Following Michael et al. (2014), we assessed the importance of each variable in terms of their scaled average weight, calculated using the models where those variables were included.

**Yearly models of WTSH fallout.**   We related the number of WTSH observed during 16 surveys of the entire study area (every three days during a single year) to the four aforementioned environmental variables: moon illumination, average wind speed, average wind direction, and Julian date (S1 File). We did not consider variable interactions, and calculated pseudo R-squared values based on the standard errors using the 'rsq' package [29].

## Results

### Fallout records

To interpret our road surveys in a broader context, we compared the number of grounded WTSH we documented along the SE corner of Oʻahu with the SLP intake records, which provided an island-wide measure of fallout timing and magnitude. The SLP intake records of fledging chicks spanned from November 2 to January 5, and our observations of grounded shearwaters along the Kalanianaʻole Highway spanned from November 6 to December 21. Overall, only 2.3% of the SLP intake records fell outside of our road survey period (November 6 –December 21), with yearly proportions ranging from 1.3% to 6.3% (S1 Table).

The total number of rescued WTSH brought into SLP yearly across the 8-year study varied by nearly an order of magnitude, ranging from 74 to 525 birds per year, with an average of 226.1 +/- 170.6 S.D. (median = 159.5) (S1 Table). The number of WTSH carcasses observed on the survey route per year also varied widely, ranging from 7 to 60 birds, with an average of 24.1 +/- 18.7 S.D. (median = 17.5) (S1 Table). There was a positive correlation between the yearly number of road-killed birds (our surveys) and rescued birds (SLP records), with 2012 and 2016 standing out as high-fallout years ($r^2 = 0.85$, df = 6, p < 0.01) (Fig 2). There were 469 rescued birds in 2012 and 525 in 2016, with both years exceeding the median by over 300 birds. Likewise, there were 60 road-killed birds in 2012 and 45 in 2016, compared to the median of 17.5 birds. The lowest numbers of rescued and road-killed birds occurred in 2018, with 74 and 7 birds respectively.

### Fallout modeling

Over the 8-year study, the number of grounded WTSH observed per survey ranged from 0 to 10, with an overall average of 1.5 +/- 2.2 S.D. (median = 1) (Fig 3). Moreover, to account for

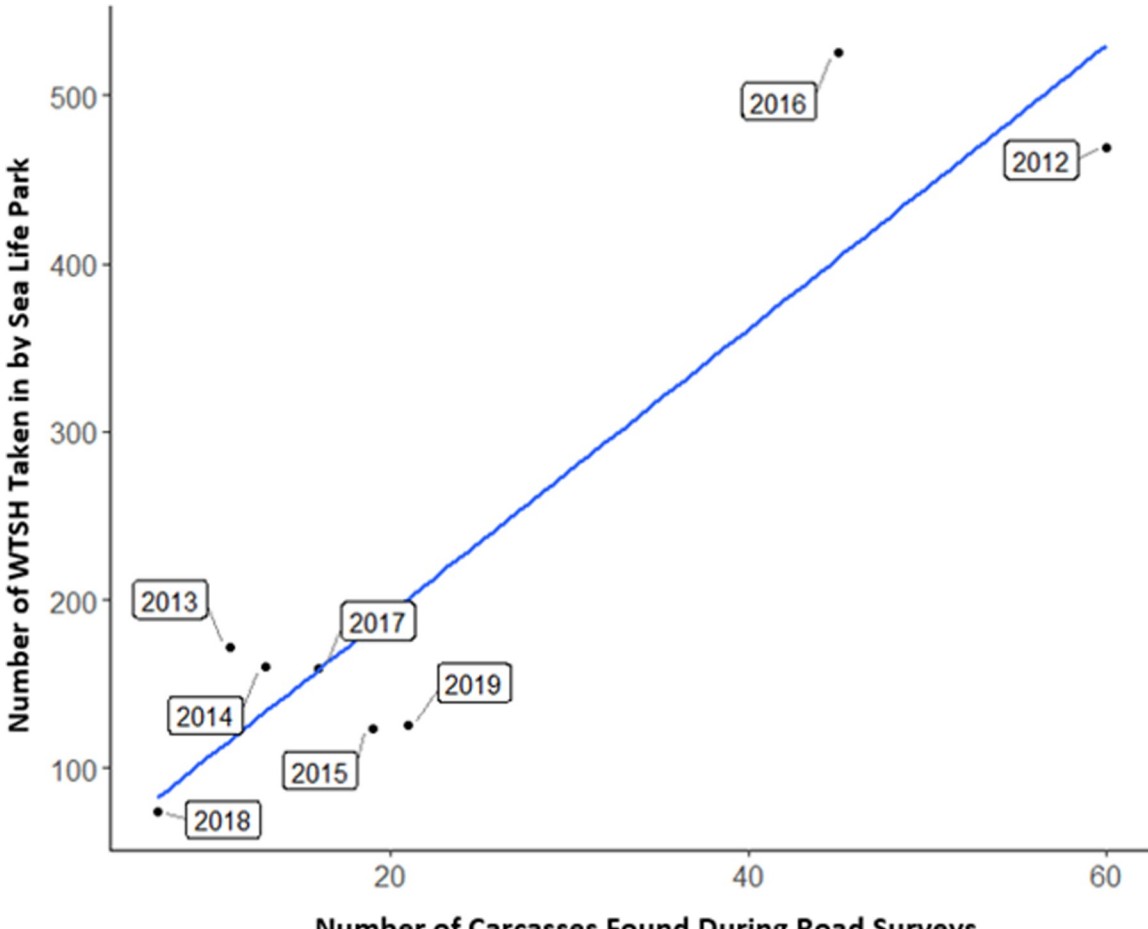

**Fig 2. Fallout comparison between Sea Life Park intake records and road surveys.** Scatterplot showing the total rescued WTSH per year from Sea Life Park intake records versus the total WTSH carcasses documented per year during road surveys ($r^2 = 0.85$).

the large proportion (46%) of absences (0 WTSH detected during a road survey), we fitted the fallout count data to Poisson (1< VMR < 2) and negative binomial (VMR > 2) distributions. We developed and fitted a full model and eight single-year models.

**Multi-year model of WTSH fallout.** Because 9 different model formulations were required to achieve an AICc weight of 0.90, this model set was used to ascertain the importance of the driver variables. Of the 9 variables tested, only the interaction between moon illumination and wind speed (moon*wind speed) achieved a scaled average weight >1 and was thus deemed an "important" variable (S2 Table). Moon, wind speed, and year all had weights of 1, because they contributed an average amount to each model's weight. Date, wind direction, (moon*date), and (wind speed*date) had weights < 1, and contributed less than the average variable to each model's weight. Light regime had a weight of 0, and did not appear in any of the models required to achieve the AICc weight of 0.90.

The overall best-fitting model had a weight of 0.37 and included four explanatory variables: moon, wind speed, year, and the interaction of moon and wind speed (moon*wind speed) (Table 1). All variables in this model were significant, except wind speed and year 2016 (not significantly different from 2012). The negative coefficient for the moon variable in this model (-2.9) indicates that, across the 8-year period, fewer birds were grounded when a greater percentage of the lunar disk was illuminated. All years except for 2016 were significantly different

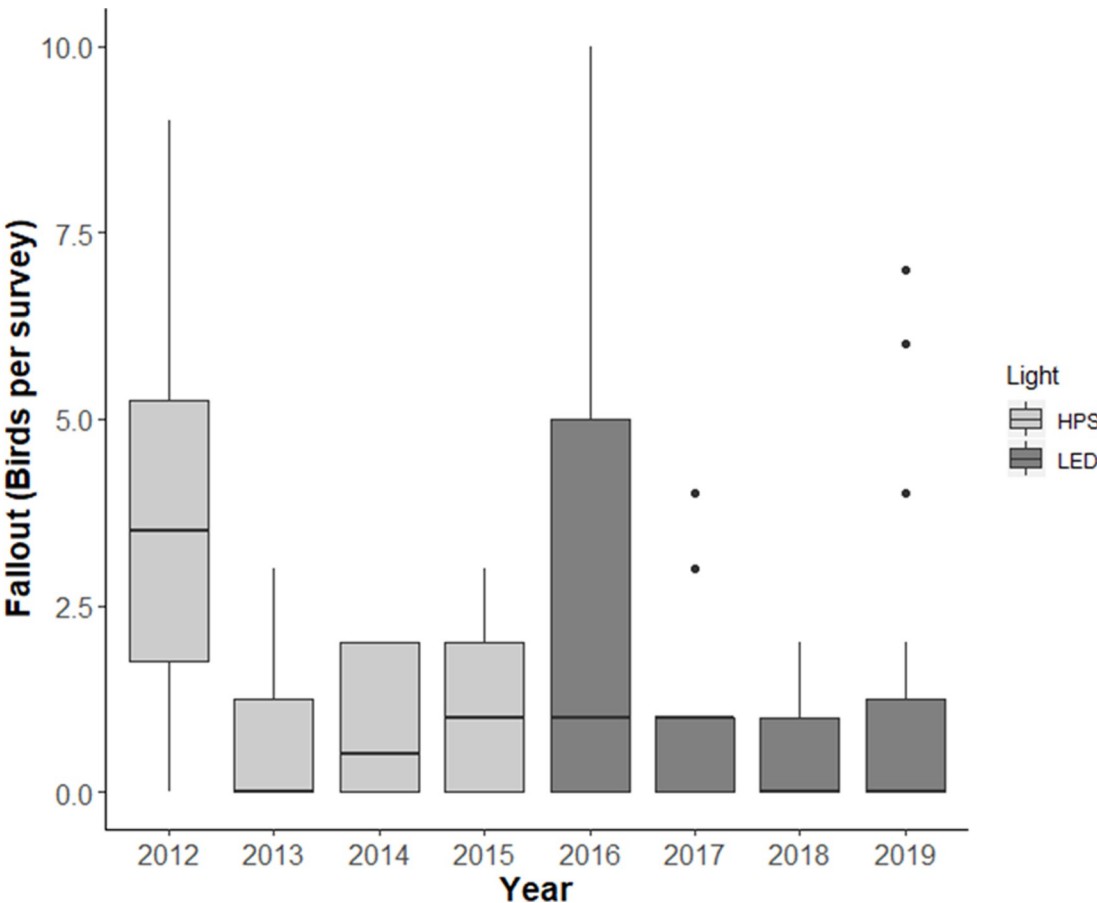

**Fig 3. Boxplots of WTSH carcasses observed during road surveys.** Distribution (5, 25, 50, 75, 95 percentiles) of the number of grounded WTSH observed each study year (n = 16 yearly surveys). Dots indicate outliers.

**Table 1. Full model output.**

| Explanatory variable | Estimate | S.E. | Z-value | p-value |
|---|---|---|---|---|
| Intercept [a] | 2.136 | 0.580 | 3.709 | **<0.001** |
| Wind Speed | -0.027 | 0.046 | -0.576 | 0.565 |
| Moon | -2.884 | 0.827 | -3.485 | **<0.001** |
| Year2013 | -1.711 | 0.447 | -3.825 | **<0.001** |
| Year2014 | -1.685 | 0.406 | -4.148 | **<0.001** |
| Year2015 | -1.490 | 0.365 | -4.082 | **<0.001** |
| Year2016 | -0.373 | 0.324 | -1.148 | 0.251 |
| Year2017 | -1.661 | 0.391 | -4.247 | **<0.001** |
| Year2018 | -2.492 | 0.483 | -5.153 | **<0.001** |
| Year2019 | -1.034 | 0.366 | -2.824 | **0.005** |
| Wind Speed*Moon | 0.225 | 0.083 | 2.710 | **0.007** |

GLM results from best-fit full model, following a negative binomial distribution. Bold font denotes significance at alpha < 0.05.

[a] Reference year (intercept) is 2012.

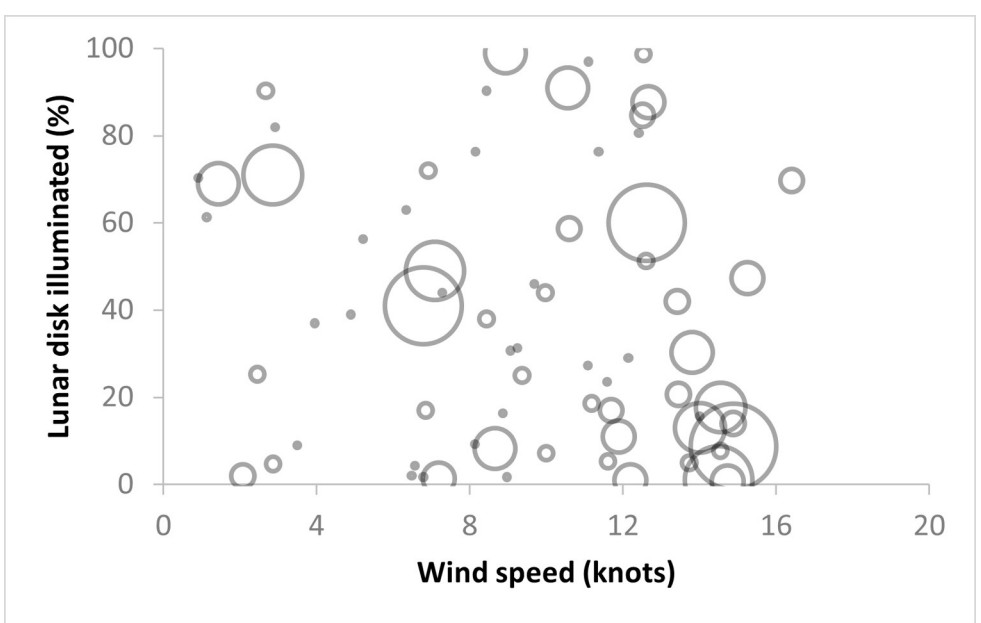

**Fig 4. Fallout as a function of moon illumination and wind speed.** Scatterplot of the number of grounded WTSH observed per survey, in relation to wind speed, and moon illumination. Open circles indicate the presence of fallout, with the increasing radius ranging from 1 to 10. Small solid dots indicate the absence of fallout (0 birds).

from the reference year (2012). The interaction between wind speed and moon had a positive coefficient (+0.23) suggesting that fallout was higher during periods of lower lunar illumination and higher wind speed (Fig 4).

**Yearly models of WTSH fallout.** In addition to the interannual variability in the number of WTSH observed during road surveys (Fig 3), the temporal aggregation of fallout across surveys was also highly variable, as evidenced by the varying dispersion (variance to mean ratio, VMR) observed yearly (1.02–4.03) (Table 2)). Moreover, due to the large proportion (46%) of absences (0 WTSH detected during a road survey), fallout counts followed a Poisson distribution ($1 < $ VMR $ < 2$) in every year, except for 2019 (VMR = 4.03), when the negative binomial model yielded a lower AICc value.

**Table 2. Yearly model output.**

| Year | Dis. | VMR | Pseudo Adj. $R^2$ | Estimate | | | | | p-value | | | | |
|------|------|-----|-------------------|----------|------|------|------|------|---------|------|------|------|------|
| | | | | Int. | WS | WD | Moon | Date | Int. | WS | WD | Moon | Date |
| 2012 | P | 2.11 | 0.41 | -29.293 | -0.744 | -0.032 | -0.065 | 0.137 | 0.096 | 0.236 | 0.408 | 0.071 | 0.054 |
| 2013 | P | 1.50 | 0.51 | -28.120 | 0.684 | 0.029 | -4.944 | 0.065 | 0.146 | **0.028** | **0.016** | **0.006** | 0.211 |
| 2014 | P | 1.02 | 0.64 | 3.055 | 0.393 | 0.003 | -1.659 | -0.020 | 0.751 | **0.045** | 0.782 | 0.144 | 0.456 |
| 2015 | P | 1.14 | 0.33 | -3.680 | 0.03 | -0.001 | 1.993 | 0.008 | 0.575 | 0.818 | 0.902 | **0.006** | 0.720 |
| 2016 | P | 3.89 | 0.50 | -0.169 | 0.268 | 0.002 | 0.439 | -0.006 | 0.979 | **0.034** | 0.816 | 0.507 | 0.777 |
| 2017 | P | 1.20 | 0.18 | 8.690 | 0.099 | -0.002 | -0.854 | -0.027 | 0.291 | 0.273 | 0.638 | 0.291 | 0.309 |
| 2018 | P | 1.20 | 0.57 | -20.160 | -0.190 | -0.001 | -1.592 | 0.064 | 0.183 | 0.546 | 0.959 | 0.261 | 0.178 |
| 2019 | NB | 4.03 | 0.26 | 42.940 | -0.100 | -0.022 | -3.305 | -0.115 | 0.056 | 0.543 | 0.196 | **0.004** | 0.055 |

GLM output of annual fallout models, based on 16 surveys (Nov. 6 –Dec. 21) and clumped data distributions (P = Poisson, NB = negative binomial), as evidenced by the variance to mean ratio (VMR). In addition to the intercept (Int.), four explanatory variables were considered: wind speed (WS), wind direction (WD), lunar illumination (Moon), and Julian Date (Date).

[a] The bold font denotes significance at alpha < 0.05. and the pseudo adjusted R-squared quantifies the model fit.

Overall, the yearly models explained a wide range of the variation in fallout throughout the fledging season, with their pseudo $R^2$ values ranging from 18% (2017) to 64% (2014). Moreover, different variables were significant in different years (Table 2). Surprisingly, the influence of moon illumination was not consistent across our study, with a significant effect in three years: it was negative twice (2013 and 2019), and it was positive once (2015). Wind speed had a significant positive effect in three years (2013, 2014, and 2016), whereby higher wind speeds led to more fallout. Wind direction had a significant positive effect once (2013), whereby wind blowing from the southwest led to more fallout. Julian date was never significant, suggesting that fallout was variable throughout the survey period (November 6 –December 21).

Overall, while fallout was explained well (pseudo $R^2 \geq 0.5$) by wind speed alone in 2014 and 2016, it was explained moderately well (pseudo $R^2 \geq 0.3$) by moon illumination alone in 2015 and 2019. In 2013, about half of the fallout variation was explained by a combination of wind speed, wind direction, and moon illumination. In three years (2012, 2017, and 2018), fallout was not significantly explained by any of the predictors.

Two years (2012 and 2016) showed significantly higher fallout compared to the other study years (Fig 3) and together accounted for 55% of the WTSH found during road surveys. Those same years were also responsible for 55% of all rescued birds brought to SLP, within the timeframe of this study. While none of the predictor variables were statistically significant in 2012, moon and date were marginally significant ($0.10 < p < 0.05$) (Table 2). The highest yearly fallout occurred in 2012, when 60 WTSH were grounded during an early new moon period (Julian days: 317–326, November 12–21), and a later one (Julian days: 344–353, December 11–18) (Fig 5), both of which were accompanied by strong winds (Fig 6). In 2016, a new moon period occurred in the middle of the fledging season, leading to a single peak in fallout (Fig 5), which coincided with a period of high wind speeds ($> 12$ knots), increasing the number of birds grounding at this time (Fig 6).

## Discussion

### Timing and magnitude of WTSH fallout

The strong positive correlation between the yearly numbers of grounded WTSH found during our road surveys and rescued WTSH brought to SLP suggests that our small-scale surveys of a fallout hotspot are indicative of island-wide fallout trends on Oʻahu. Both the rescue records and the road surveys documented the highest fallout in 2012 and 2016, and the lowest fallout in 2018. Moreover, only 2.3% of the WTSH brought to Sea Life Park during the fledging season between 2012–2019 fell outside of our study period (November 6 –December 21), suggesting that our survey window captures most of the fledging season fallout.

### Interpretation of model results

**Multi-year model of WTSH fallout.** Our hypothesis that the LED streetlights would increase shearwater groundings due to higher sensitivity to shorter wavelengths was not supported, as the light regime was not selected as a significant predictor variable in any of the top models. It is possible that shearwater visual perception of LED lights was in fact greater, but shielding reduced initial attraction, thus balancing out overall fallout. However, even if this were the case, our analysis could not distinguish between these two factors, because the changes in bulb type and shielding were not independent. Nonetheless, this finding has useful implications for resource managers since LED lights are a common replacement for HPS lights in Hawaiʻi and elsewhere. While we encourage managers to seek lighting adjustments that will mitigate fallout, our study shows that the change in streetlights from unshielded HPS to shielded 3000 K– 4000 K LED did not exacerbate this problem for WTSH on Oʻahu.

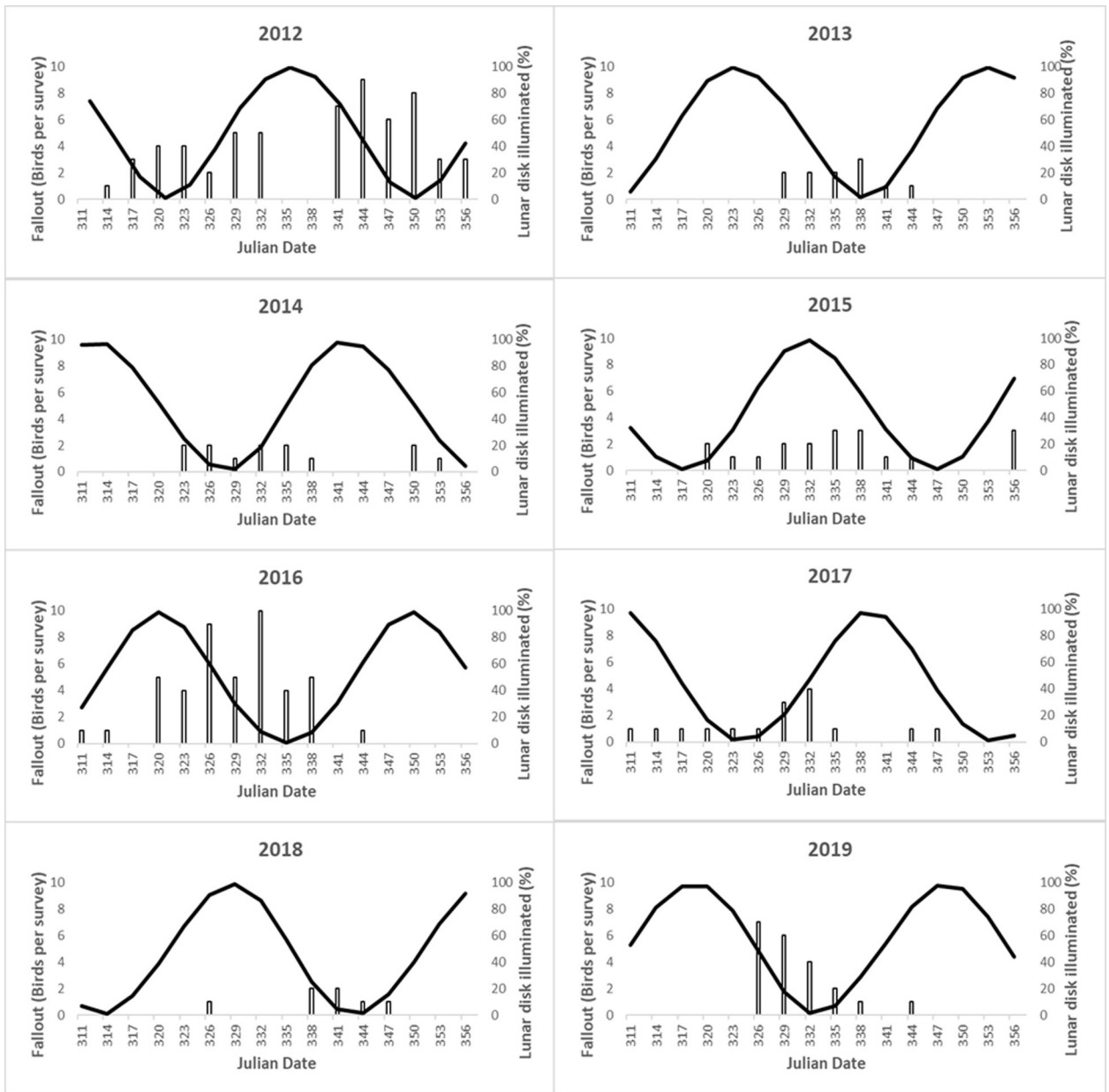

**Fig 5. Moon illumination and fallout.** Time series of the number of grounded WTSH observed per survey (bars) and nightly lunar illumination (back line). The dates on the x-axis indicate the survey days.

It is possible that, even if there was an effect of light type on WTSH fallout, its influence was marginal compared to the effect of the other environmental drivers. In particular, the higher than average variable weight of the interaction between moon and wind speed suggests that fallout is a dynamic process, driven by the synergy of low moon illumination and strong winds, more so than by moon or wind alone (Fig 4, S2 Table). While previous studies have identified the importance of moon and wind, this is the first time their interaction has been considered.

This significant interaction underscores a conceptual model, whereby wind speed determines the magnitude of fledging birds departing their colonies, and the moon illumination

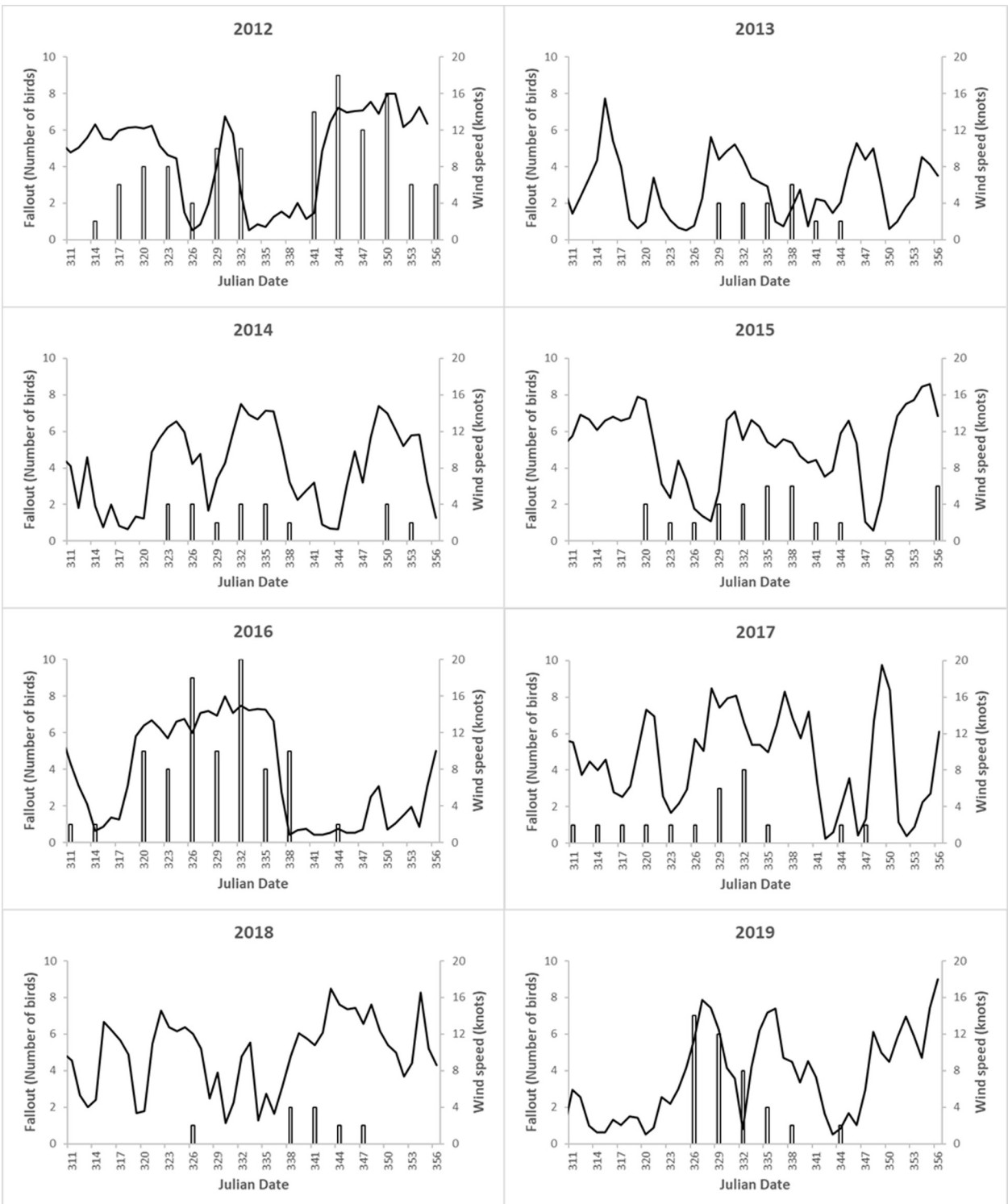

**Fig 6. Wind speed and fallout.** Time series of the number of grounded WTSH observed per survey (bars) and nightly wind speed (back line). The dates on the x-axis indicate the survey days.

determines the attraction of those fledglings toward onshore lighting. This conceptual model can explain why years like 2016, when the peak of the fledging season coincided with a new moon and high wind speeds, have greater fallout.

While the interaction of the lunar illumination and the wind speed was critical, the timing of these variables, captured using their interactions with date, were less important. Together, these results suggest that, within the time frame of our study, fallout is most dependent on the temporal overlap of low moon illumination and high winds, rather than on their specific timing.

**Yearly models of WTSH fallout.** The influence of the four predictor variables (wind speed, wind direction, moon, and Julian date) were not consistent across all 8 study years. While most previous studies have found strong negative relationships between moon illumination and fallout, our yearly models only documented this pattern in 2 years [4, 8, 10, 11, 33]. One possible explanation for this result could be a mismatch between moonrise / moonset times and WTSH fledging. We used an average lunar disk illumination for the three nights prior to each survey, assuming that this would be representative of visible moonlight while WTSH were fledging. However, if most birds fledge shortly after sunset, before the moon rises during waning moon phases, conditions will resemble a new moon [16, 34]. Similarly, if island topography or clouds obscure a rising moon from a given natal colony, the navigational benefits provided by the lunar disk could be compromised until the moon rises over obscuring landscape features. The peak fledging times of WTSH are unknown but could be useful to improve our understanding of the influence of moonrise / moonset times on fallout. Furthermore, the lack of strong lunar trends in the yearly models could be due to small sample sizes, with each year only involving 16 surveys.

Contrary to Rodríguez et al. (2014), who documented a significant increase in fallout as the fledging season progressed, our yearly models did not find a date effect. However, the likelihood of finding an effect of date depends partly on the timing and the duration of the study period. While opportunistic studies using intake records and citizen-science programs sample a wider temporal window, spanning before and after the fledging season, our surveys spanned a narrow temporal window during the WTSH fledging season. Thus, our results suggest that, due to interannual variability in the timing and magnitude of fallout, on average it is distributed evenly throughout our study period (November 6 –December 21). Fledging primarily occurs during this 6-week period, and is likely modulated by a variety of factors, including breeding phenology, chick development, and environmental conditions [10, 33, 35].

The positive relationship between wind speed and fallout is likely related to the fledglings using this environmental cue for fledging and relying on wind to take flight. One possible explanation for why we observed more fallout with higher windspeeds is that intermediate to strong windspeed enable WTSH to take flight, however a lack of flight experience and muscle development may make it difficult for fledgling birds to navigate in strong winds, thus leading to more fallout under higher wind speeds [36, 37].

Although the model results suggest that wind speed is more important than wind direction, an exception to this general pattern was observed in 2013, when peak fallout coincided with a period of moderate to weak southerly winds (S1 Fig). A previous study on Oʻahu suggested that winds from the southeast were more common during a year of very high fallout and hypothesized that birds were advected to the northwest and deposited along the entire windward coast of the island [22]. The lack of significance of wind direction in years other than 2013 may be related to the prevailing wind patterns on windward Oʻahu, which rarely switch from the northeast direction, thus limiting the comparison of different wind directions. Furthermore, because south-westerly winds during the study are characterized by lower speeds, any potential influence of direction is not independent from the wind speed effect discussed previously (S2 Fig).

Previous studies show an increase in fallout when prevailing winds are directed toward brightly lit coastal areas [10, 11]. Yet, the influence of wind direction is difficult to interpret since headings are circular (0–360 degrees) and should be carefully considered on a case-by-case basis. Because the prevailing winds in our study area are the northeasterly trade winds or southern (Kona) storms, these bi-directional wind headings facilitated the analysis and simplified the interpretation. With the exception of 2013, peak fallout occurred during trade winds (Fig 6).

Although our study did not show a significant effect of Julian date on fallout, previous road surveys from Oʻahu spanning ten years (2002 to 2010), revealed that November 25 was the peak fallout date, with 67% of the grounded WTSH found during a one-week period (21–27 November) [21]. We hypothesize that the timing of the moon phase, in relation to this peak fallout period could explain the interannual relationship of moon illumination and fallout. Namely, higher fallout occurs in years when the new moon overlaps the peak fallout week (21–27 November).

In 2015, the full moon occurred on November 28, whereas in 2013 and 2019 the moon phase was closer to a new moon on that date. It appears that when peak fallout coincides with a new moon, a single fallout peak occurs, thus causing a negative correlation with moon illumination. However, if peak fallout coincides with a full moon, the unimodal pattern breaks down, resulting in two smaller fallout peaks. Previous work yielded a quadratic relationship between the timing of the full moon and the number of Newell's shearwater (*Puffinus newelli*) fallout, with fewer total groundings when the full moon occurred during the middle of the month [5]. When we replicated this analysis for WTSH, the quadratic model was not significant ($R^2$ = 0.038, $F_{2,5}$ = 1.140, p = 0.39), suggesting that annual fallout did not follow the same pattern with the timing of the full moon. Although, other variables such as the timing of moon rise, cloud cover, and topography blocking the moon were not taken into account and may play a role in the moon's influence on fallout. Demographic factors, involving the size of the breeding population and the reproductive success likely influence the yearly supply of fledging chicks [5].

## Implications for fallout mitigation

Our results are reassuring because they suggest that the shielded LED streetlights did not increase WTSH mortality due to fallout, as we hypothesized. Given the strong correlation between the dead birds observed in our road surveys and the live birds brought to SLP, there is no evidence suggesting that the shielded LED streetlights impacted the number of birds affected by fallout overall. However, because these new lights did not reduce fallout, wildlife managers may consider modifications such as dimming, wavelength alteration or motion sensors, to mitigate negative impacts to fledging WTSH on Oʻahu [6, 19].

A recent survey of lighting experts suggests that while LEDs can be adjusted to reduce light pollution and minimize wildlife impacts, yet municipalities rarely capitalize on those benefits [19]. For instance, although new-technology LED streetlights can filter out lower wavelengths [17], full spectrum white LED lights maximize brightness, and are commonly chosen to replace HPS streetlights. Furthermore, LEDs come in a variety of CCTs with options as low as 2200 K, the maximum temperature experts recommend for wildlife [17]. However, municipalities commonly implement 3000–5000 K LED streetlights because of their efficiency for human use [19]. Future studies should compare different LED lighting options in areas where seabird fallout occurs to determine the characteristics that best mitigate negative impacts to seabirds and other wildlife.

While it may be unfeasible to reduce light pollution wherever fallout occurs, areas near breeding colonies could be targeted for localized management [21]. In addition to diminishing

light pollution during the fledging season, we also encourage community-based rescue efforts for WTSH to target fallout hotspots on Oʻahu on nights with low moon illumination and strong winds. Further documentation of fallout hotspots could help guide lighting management and rescue efforts throughout the Hawaiian Islands.

Finally, predictive fallout models are limited by the lack of comprehensive annual population estimates, which might have explained some of the interannual variation in the number of grounded birds. Thus, annual WTSH breeding population sizes and reproductive success would likely improve our understanding of fallout interannual variability and trends in Hawaiʻi. The findings and conclusions in this article are those of the authors and do not necessarily represent the official views of the U.S. Fish and Wildlife Service.

## Supporting information

**S1 Table. Annual data from sea life park and road surveys.** Comparison of annual WTSH fallout magnitude (total number of grounded birds) and timing (date ranges) from Sea Life Park intake records and road surveys (this study). Summary statistics (mean, median, and range) refer to the number of grounded birds encountered yearly, based on 16 standardized surveys spanning November 6 to December 21.
(TIF)

**S2 Table. Variable importance in AICc analysis.** Scaled average variable weights. (>1 values indicate greater than average weight when variable was included in model; weights = 1 are average, weights <1 less than average).
(TIF)

**S1 Fig. Wind direction and fallout in 2013.** Wind direction and fallout during the 2013 fledging season. Black line is wind direction and white bars are number of birds per survey.
(TIF)

**S2 Fig. Wind speed and wind direction.** Scatterplot of wind speed and wind direction during the fledging seasons 2012–2019 ($R^2$ = 0.71).
(TIF)

**S1 File. WTSH fallout data from road surveys.** Data from road surveys (total = 128) including variables year, Julian date, moon illumination (%), wind speed (knots), wind direction (degrees), light regime (HPS = high pressure sodium, LED = light emitting diode), and number of grounded WTSH observed.
(XLSX)

## Acknowledgments

We would like to thank Jeff Pawloski for providing Sea Life Park intake records, David Field and Susan Carstenn for input on the analysis and writing, and Pelagicos lab members who assisted with the road surveys: Sarah Donahue, Michelle Hester, Angelica Moua, Anessa Musgrove, and Dan Rapp.

## Author Contributions

**Conceptualization:** K. David Hyrenbach, Keith Swindle.

**Data curation:** Jennifer Urmston.

**Formal analysis:** Jennifer Urmston.

**Funding acquisition:** K. David Hyrenbach.

**Investigation:** Jennifer Urmston.

**Methodology:** Jennifer Urmston, K. David Hyrenbach, Keith Swindle.

**Project administration:** K. David Hyrenbach.

**Supervision:** K. David Hyrenbach.

**Visualization:** Keith Swindle.

**Writing – original draft:** Jennifer Urmston.

**Writing – review & editing:** K. David Hyrenbach, Keith Swindle.

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
