## [Decision Letter · Decision Letter 0]

1 Nov 2021

PONE-D-21-31078Timing and Magnitude of Wedge-tailed Shearwater(Ardenna pacifica) Fallout on Southeast O'ahu, Hawaiʻi:

A Dynamic Interaction Between Moon and WindPLOS ONE

Dear Dr. Urmston,

Thank you for submitting your manuscript to PLOS ONE. After careful consideration, we feel that it has merit but does not fully meet PLOS ONE’s publication criteria as it currently stands. Therefore, we invite you to submit a revised version of the manuscript that addresses the points raised during the review process.

We look forward to receiving your revised manuscript.

Kind regards,

Vitor Hugo Rodrigues Paiva, Ph.D.

Academic Editor

PLOS ONE

Journal Requirements:

“This work was supported by Experiment.com (Blinded by the light: reducing shearwater deaths along a coastal highway in O 'ahu, Hawai'i) and The Eppley Foundation for Research (Blinded by the Light: Shearwater Deaths Along a Coastal Highway in O'ahu)”

4. We note that Figure 1 in your submission contain map images which may be copyrighted. All PLOS content is published under the Creative Commons Attribution License (CC BY 4.0), which means that the manuscript, images, and Supporting Information files will be freely available online, and any third party is permitted to access, download, copy, distribute, and use these materials in any way, even commercially, with proper attribution. For these reasons, we cannot publish previously copyrighted maps or satellite images created using proprietary data, such as Google software (Google Maps, Street View, and Earth). For more information, see our copyright guidelines: http://journals.plos.org/plosone/s/licenses-and-copyright.

5. Please include a copy of Table A1 which you refer to in your text on page 11.

6. Please upload a copy of Supporting Information Figure S1 and S2 which you refer to in your text on page 18 and 19.

Reviewers' comments:

Reviewer's Responses to Questions

**Comments to the Author**

1. Is the manuscript technically sound, and do the data support the conclusions?

Reviewer #1: Yes

Reviewer #2: Yes

Reviewer #3: Yes

2. Has the statistical analysis been performed appropriately and rigorously? 

Reviewer #1: I Don't Know

Reviewer #2: Yes

Reviewer #3: Yes

3. Have the authors made all data underlying the findings in their manuscript fully available?

Reviewer #1: Yes

Reviewer #2: Yes

Reviewer #3: Yes

4. Is the manuscript presented in an intelligible fashion and written in standard English?

Reviewer #1: Yes

Reviewer #2: Yes

Reviewer #3: Yes

5. Review Comments to the Author

Reviewer #1: This manuscript reveals a novel analysis of the effect of moon, wind speed and wind direction on the fallout of wedge-tailed shearwaters. Additionally, there was a transition from high-pressure sodium lamps to LED lamps on the route of the regular fallout survey. This provided a unique opportunity to study the effect of the change in streetlamps on the number of grounded birds. The study reveals that while the transition from HPS to LED streetlamps did not influence the fallout, the interaction between moon and wind speed did. While the data and analysis provide a new insight into the grounding of burrow-nesting shearwaters, I believe that the manuscript itself could be improved. Firstly, I think that moving sections of full model before the yearly models could improve the focus of the study. Furthermore, the authors could consider including the timing of moonrise, moonset, as well as cloud cover in their model. Finally, I include minor points in the attachment that could improve clarity and readability of the manuscript. My comments include first words of each line I referred to, since no line numbers were provided.

Reviewer #2: Overall it is an important paper that provides further evidence for effects of drivers in fallout events, crucially it provides initial evidence for the effect of altering urban lighting.

The text is very well written and clear without the need for much correction. I found the methods section too detailed and would suggest compressing some information and maybe place it in a supplement material annex.

Both the results and the discussion highlight the environmental aspects of the analysis while the LED is nearly absent from the results and, albeit well discussed, does not have a prominent placement in the discussion. I would suggest rearranging the paper in a way that highlights the LED vs HPS change as this is a most crucial information for current light pollution mitigation efforts. I would also suggest that the authors increase the analysis of the data by including if possible the total fallout per year registered by the SLP which correspond to the survey area, thus increasing the scope of light regime change analysis. You provide the totals in Fig.2 it would be possible to add this data to the light regime comparison? I understand that by drawing a parallel between SLP fallout records and your survey dead birds you are showing that your data is potentially indicative of the overall fallout and I agree, however I feel the analysis and results feel flat and could be worked on to present a more robust evidence.

Personally I would also suggest changing the title, moon and wind effects on fallout are known and not novel, their interaction is expected, on the other hand LEDs effects on fallout are unknown, understudied and needed!

As a final note the authors should take extra care when presenting manuscript for assessment, you are missing Figure 4, the lines are not numbered and the table was outside the page bounds so unreadable in the pdf version. I added line to the word version so that my comments were easier to follow.

Major changes:

> the authors collected data on utility poles (lighting systems) nearest the grounded bird. They could present an analysis of this data, i.e. was it possible to identify specific areas within the transects or fallout was widespread across it? Does this coincide with the previous research of Friswold et al 2020 (so data from 2002-2010) ? it could be interesting to discuss the implications of, after the change in lighting systems, the locations for fallout remain the same or change, especially in relation to the colonies you identify in line 138-142.

> The authors only used data of dead birds. Were live birds observed during the transects and if so were they added to the rescue center tally? I miss some discussion regarding the dead versus live birds in relation to the surveys. I understand that the authors provided a parallel between the two datasets by comparing proportions of live (rescue center data) and dead (this study), but if possible it would be interesting to include the live rescued birds in the full models.

For instances the change to LED did not provide increased deaths however is it possible to evaluate its effect on the total fallout numbers provided by the rescue center? I understand that it might not be possible to confirm the location of fallout for all birds and that the rescue center possibly obtains birds from outside the survey area, however if it were possible to include the birds that have been rescued within the survey area it would greatly improve the results and further increase the impact of this work. Thus discussion not only the overall negative effect of the light pollution (fallout) and the ‘no-effect’ of the light change as well as the mortality associated with both.

> Figure 4 is missing. If possible maybe combine the two time series graph (I understand this will generate three y scales but perhaps there is a way to illustrate the moon illumination as well).

> reduce the size of paragraph 5 in the introduction

> methodology could be shortened and the information placed in a supplementary material, for example the passages pertaining to the specificity of Poisson distributions and the AIC (lines 192-195; 211-221)

> If possible include 95% Confidence intervals in the results from your models. I fell it would provide a more cohesive interpretation of the results.

Minor changes:

I have added line numbers to the text version of the ms as they were missing.

> Carefull with the use of abbreviations such as HDOT. PLOS ONE guidelines state ‘Do not use non-standard abbreviations unless they appear at least three times in the text.’

> in the CCT mentions throughout the text, remove the degree symbol before K (absolute temperature Kelvin is not used with the degree symbol)

> line 43. Remove comma after petrels: ‘petrels and puffins’

> line 50. ‘conditions’ is repeated

> line 60. Add recent reference regarding powerline collisions Travers et al. 2021 Avian Conservation Ecology

> line 79. I disagree that a CCT >2700K is high. I would substitute by ‘recommended’ for example. Organism friendly.

> line 292 and 293. Revise text. Two sentences are unconnected, ‘yet’ followed by ‘however’ is confusing.

> Table 1. Correct to ‘mean ratio (VMR). In’

> Figure 3. Perhaps it would be appropriate to add a line plot with total of rescue birds from the rescue centre.

> Figure 5. I find this graph particularly useful for the wind and moon integrated evaluation. The smaller circles represent 0 fallout? I would substitute all 0 fallout records with a different symbol, for example an ‘x’ or a ‘+’. As it is its not possible to discern surveys with no fallout or surveys with low fallout. I wonder if using two colors to represent the two light regimes would also be usefull in this graph.

Reviewer #3: This opportunistic study provides much needed information on how changing light characteristics influences (or in this case, failed to influence) stranding rates of shearwaters vulnerable to fallouts, in addition to investigating more thoroughly the influence of environmental conditions, including moon phase and wind conditions. The manuscript is very clearly written and well laid out. The statistical analyses appear sound (however, please note that this is not my area of expertise) and results are clearly reported, followed by a well-referenced and interesting discussion. Overall, I very much enjoyed reading this manuscript and only have minor edits, as follows:

Page 4, last paragraph: “wedge-tailed shearwaters” should be capitalized.

Page 5, second line: USFWS should be spelled out the first time this acronym is used.

Page 6, second paragraph, last sentence: The prediction that northeasterly winds would lead to higher fallout rates goes against the observation that more birds were found stranded during anomalous southerly winds in 1994, as stated on page 5, second paragraph. Please clarify. Also, it would be useful to add an explanation of why >8 mph winds was chosen as “strong”. For example, in other parts of the world, winds under 13 mph would be classified as low, so it would be good for the reader to have a better understanding of what typical wind speeds are for this particular region. Also, for consistency, it would be better to present all wind speeds as knots, rather than a combination of mph and knots.

Table 1: the p-values for moon and date were cut-off from the table in the produced document.

While the Figure 4 caption is present in the text, the actual figure is missing from the document.

6. PLOS authors have the option to publish the peer review history of their article (what does this mean?). If published, this will include your full peer review and any attached files.

Reviewer #1: **Yes: **Martyna Syposz

Reviewer #2: No

Reviewer #3: No

---

## [Author Response · Author response to Decision Letter 0]

28 Dec 2021

RESPONSES TO REVIEWER COMMENTS

Journal Requirements

Comment 1: Please ensure that your manuscript meets PLOS ONE's style requirements, including those for file naming. The PLOS ONE style templates can be found at

Response: We have formatted the ms according to the journal requirements. 

Comment 2: Thank you for stating the following financial disclosure:

“This work was supported by Experiment.com (Blinded by the light: reducing shearwater deaths along a coastal highway in O 'ahu, Hawai'i) and The Eppley Foundation for Research (Blinded by the Light: Shearwater Deaths Along a Coastal Highway in O'ahu)”

Response: We provided an amended Role of Funder statement in our cover letter, to reflect that this work was supported by Experiment.com (Blinded by the light: reducing shearwater deaths along a coastal highway in O 'ahu, Hawai'i) and The Eppley Foundation for Research (Blinded by the Light: Shearwater Deaths Along a Coastal Highway in O'ahu). The funders had no role in study design, data collection and analysis, decision to publish, or preparation of the manuscript.

Comment 3: In your Data Availability statement, you have not specified where the minimal data set underlying the results described in your manuscript can be found. PLOS defines a study's minimal data set as the underlying data used to reach the conclusions drawn in the manuscript and any additional data required to replicate the reported study findings in their entirety. All PLOS journals require that the minimal data set be made fully available. For more information about our data policy, please see http://journals.plos.org/plosone/s/data-availability.

Response: We provided a data file for inclusion as supplementary table 2, which includes all road survey observations and environmental data, sampled every three days between Nov. 6 and Dec. 21 for eight years (2012 – 2019). These were the data used to develop the multi-year full model and the eight single-year yearly models. 

Comment 4: We note that Figure 1 in your submission contain map images which may be copyrighted. All PLOS content is published under the Creative Commons Attribution License (CC BY 4.0), which means that the manuscript, images, and Supporting Information files will be freely available online, and any third party is permitted to access, download, copy, distribute, and use these materials in any way, even commercially, with proper attribution. For these reasons, we cannot publish previously copyrighted maps or satellite images created using proprietary data, such as Google software (Google Maps, Street View, and Earth). For more information, see our copyright guidelines: http://journals.plos.org/plosone/s/licenses-and-copyright.

Response: Figure 1 has not been previously published elsewhere but is an original created using Arc GIS Pro. This figure shows a series of geographic features overlaid on the ArcGIS Pro Software Version 2.5 topographic basemap (ESRI 2020). To this end, we added a citation to the references section: 

ESRI (2020). ArcGIS Pro (Version 2.5). Esri Inc. https://www.esri.com/en-us/arcgis/products/arcgis-pro/overview. 

Comment 5: Please include a copy of Table A1 which you refer to in your text on page 11.

Response: We apologize for this oversight. We added this table, relabeled as Table S1, in the Supplementary materials at end of manuscript. 

Comment 6: Please upload a copy of Supporting Information Figure S1 and S2 which you refer to in your text on page 18 and 19.

Response: We apologize for this oversight. We have uploaded Figures S1 and S2, in the Supplementary materials at end of manuscript. 

Comments from Reviewer 1 

Comment 1: Firstly, I think that moving sections of full model before the yearly models could improve the focus of the study.

Response: Thank you for the suggestion. We reorganized the ms and now focus on the multi-year model, which addresses the shift from the HPS to the LED lights. Then, we discuss the year-to-year variability using the single-year models. 

Comment 2: Furthermore, the authors could consider including the timing of moonrise, moonset, as well as cloud cover 

Response: While we agree that developing a more comprehensive metric if lunar illumination that includes the timing of moonrise / moonset and the cloud cover could improve the fit of the model, we chose to not do so for two reasons: (1) complexity of weather conditions and (2) applicability of the model results. Thus, we respectfully would prefer to not modify the current environmental datasets and resulting models. 

It is inherently difficult to obtain a cloud cover metric that accurately describes our study site, which encompasses a section of the windward and a section of the leeward coasts of our island. Because storms sweep across from the north or from the south, the weather on the windward and leeward shores of our island is often different. Thus, cloudiness and precipitation are often very different on either side (northwest or southeast) of Makapuu Point. Thus, developing a metric for the entire survey would require a weighing the average of the cloud cover on both shores, using the relative survey length on along shores. While this approach is doable, we fear that it would hinder the applicability of our model results, for resource managers and wildlife rescue organizations. 

Thus, to ensure the wide use of our model results, we decided to used “simple” environmental variables available from internet datasets: percent moon illumination from the US Naval Observatory and hight-time (6 pm to 6 am) hourly wind speed from the Pacific Islands Ocean Observing System. Using these variables will allow managers and wildlife rescuers to develop predictive models using wind data predictions and the lunar almanac. 

Comment 3: Finally, I include minor points in the attachment that could improve clarity and readability of the manuscript

Accepted most minor edits and made changes accordingly in manuscript. 

Comments from Reviewer 2

Comment 1: I found the methods section too detailed and would suggest compressing some information and maybe place it in a supplement material annex

Response: We reorganized the methods section and moved some of the material to a “Fallout Modelling” section, to minimize redundancy and streamline the methodological descriptions. While we did not manage to shorten this section substantially, we feel this information is useful so readers can fully evaluate the data analysis. Thus, we would like to retain this material in the ms, rather than moving it to the supplementary materials section. 

Comment 2: Both the results and the discussion highlight the environmental aspects of the analysis while the LED is nearly absent from the results and, albeit well discussed, does not have a prominent placement in the discussion. I would suggest rearranging the paper in a way that highlights the LED vs HPS change as this is a most crucial information for current light pollution mitigation efforts.

Response: Originally, we focused the ms on the environmental drivers of fallout, rather than on the influence of the changes in highway lighting, because the statistics highlighted the significant influences of moonlight and wind and did not find an overall difference before / after the changes in lighting regimes. Yet, given the reviewer’s suggestions we have refocused the ms to highlight the analysis of the lighting regimes in two ways: (i) we have modified the title of the ms, and (ii) we have rearranged the material in the methods / results / discussion sections, by placing the multi-year model before we discuss the yearly models. 

Comment 3: I would also suggest that the authors increase the analysis of the data by including if possible the total fallout per year registered by the SLP which correspond to the survey area, thus increasing the scope of light regime change analysis. You provide the totals in Fig.2 it would be possible to add this data to the light regime comparison? I understand that by drawing a parallel between SLP fallout records and your survey dead birds you are showing that your data is potentially indicative of the overall fallout and I agree, however I feel the analysis and results feel flat and could be worked on to present a more robust evidence.

Response: While analyzing the SLP intake fallout records would be a valuable and insightful exercise, we feel that this analysis is not appropriate in our current ms for three reasons: 

First, there is a spatial mis-match of the two datasets: our surveys focus on a fallout hotspot within a relatively small geographic area, and the SLP intake records include island-wide returns of grounded birds. Moreover, because most of the grounded birds do not include detailed location information, usually merely the city or highway where they were found, it would be impossible for us to subset these records to only analyze returns from our survey area. In fact, 20 – 40 of the records in any given year, do not include any location data. Moreover, while we know when the lighting regime changes for our survey area occurred between spring and summer of 2016, the shift from HPS to LEDs has been a lengthy process, with different areas of the island transitioning at different times over the span of a 2-year period (2015 – 2017). Thus, we would not be able to break up the SLP time series into a HPS vs an LED lighting regime. These uncertainties would greatly complicate the analysis of fallout trends.

Second, there is an environmental driver mismatch: due to the different spatial extent and the timing (daily vs every 3 days) of the two datasets, we would have to either perform the SLP analysis using daily environmental data or we would have to aggregate the SLP data into 3-days periods to match the timing of the road surveys. The former approach would complicate the interpretation of the two separate analyses, and the latter approach would miss the fine temporal resolution in the SLP data.

Finally, due to these inherent mis-matches, we are concerned that including both analyses in this ms would greatly blur the main messages of the ms, because we would be discussing and comparing four analyses: two multi-year models (SLP intake and road surveys) and two sets of yearly models (SLP intake and road surveys). If the model results agreed, it would still be difficult for the readers to digest all of these results. If the model did not agree, then the discussion section of the ms would have to discuss the reasons behind these discrepancies. 

Thus, we would like to only include the highly-standardized and focused road surveys in this ms. Once we have shown that the change in highway lighting did not have an effect, future papers can investigate island-wide fallout patterns using a larger dataset (2012 – 2021).

Comment 4: Personally I would also suggest changing the title, moon and wind effects on fallout are known and not novel, their interaction is expected, on the other hand LEDs effects on fallout are unknown, understudied and needed!

Response: Following this reviewer’s suggestion, we have changed the title of the ms to reflect the influence of changing lighting conditions, rather than the environmental drivers (moon and wind). However, because two changes in lighting conditions took place simultaneously: light bulb (HPS to LED) and light fixture design (unshielded to shielded), we do not explicitly mention LEDs in the title. Thus, our revised title states: “Quantifying wedge-tailed shearwater (Ardenna pacifica) fallout after changes in highway lighting on Southeast Oʻahu, Hawaiʻi”

Comment 5: The authors collected data on utility poles (lighting systems) nearest the grounded bird. They could present an analysis of this data, i.e. was it possible to identify specific areas within the transects or fallout was widespread across it? Does this coincide with the previous research of Friswold et al 2020 (so data from 2002-2010) ? it could be interesting to discuss the implications of, after the change in lighting systems, the locations for fallout remain the same or change, especially in relation to the colonies you identify in line 138-142.

Response: We absolutely agree with this comment! In fact, after investigating the influence of the lighting regimes on overall fallout first, we performed a detailed spatial analysis of these fallout data. We chose to focus this first ms on the lighting regime and will publish the fine-scale spatial analyses on a second ms. We felt that publishing both analyses together would yield a rather long and unfocused ms. 

Comment 6: The authors only used data of dead birds. Were live birds observed during the transects and if so were they added to the rescue center tally? I miss some discussion regarding the dead versus live birds in relation to the surveys. I understand that the authors provided a parallel between the two datasets by comparing proportions of live (rescue center data) and dead (this study), but if possible it would be interesting to include the live rescued birds in the full models. For instances the change to LED did not provide increased deaths however is it possible to evaluate its effect on the total fallout numbers provided by the rescue center? I understand that it might not be possible to confirm the location of fallout for all birds and that the rescue center possibly obtains birds from outside the survey area, however if it were possible to include the birds that have been rescued within the survey area it would greatly improve the results and further increase the impact of this work. Thus discussion not only the overall negative effect of the light pollution (fallout) and the ‘no-effect’ of the light change as well as the mortality associated with both.

Response: First of all, we would like to clarify that our surveys involve both live and dead birds. Whenever we found live birds along the road, we recorded them for our survey and brought them to Sea Life Park. Yet, the vast majority (99%) of the WTSH we encounter during our road surveys are dead, because by the time our surveys occur in the morning, most of the grounded shearwaters have either made their way off the road or have been hit by cars. 

Regarding the suggestion to include the SLP intake records of rescued birds in the analysis, we would like to keep these two distinct datasets apart due to their inherently different spatial / temporal scales. Please see previous explanation in response to a similar suggestion by reviewer #1. 

The reasons why we included the SLP intake records in this paper were to: (i) show that our surveys from Nov 6 to Dec 21 sampled the period of fallout throughout the island; and (ii) show that – despite the small geographic scale of our study, the year-to-year variability in fallout we documented was correlated to the pattern of island-wide fallout. Because there was a significant cross-correlation of both datasets (which shared 85% of their year-to-year variability), we contend that the same interannual drivers (e.g., ocean productivity, weather patterns) of island-wide fallout also influence the magnitude of fallout within the hotspot in the SE corner of the island. 

Comment 7: Figure 4 is missing. If possible maybe combine the two time series graph (I understand this will generate three y scales but perhaps there is a way to illustrate the moon illumination as well).

Response: Apologies for erroneously leaving out Figure 4. I have added it back in. I had considered combining the two time series, and attempted it, but felt that it made the graph difficult to read and interpret, thus decided to keep them separate. 

Comment 8: reduce the size of paragraph 5 in the introduction

Response: We removed some text and condensed the introduction.

Comment 9: methodology could be shortened and the information placed in a supplementary material, for example the passages pertaining to the specificity of Poisson distributions and the AIC (lines 192-195; 211-221)

Response: We removed some text and reorganized this section, which includes a general opening and then splits into two modelling sections (multi-year vs yearly models). This new organization removed some redundancy

Comment 10: If possible include 95% Confidence intervals in the results from your models. I fell it would provide a more cohesive interpretation of the results.

Response: While we see how adding 95% confidence intervals may enhance the interpretation of the results, we included merely the parameters and associated values for two reasons: 1) logistically, we wanted to keep the table small ensure that it was not overwhelming and 2) statistically, because the yearly models are based on small sample sizes (n=16), they provide an exploratory perspective. Thus, we want to de-emphasize the specific parameter values. Rather we only provide the parameter estimates to show the +/- signs and the p-values for reference. If we added the SE’s we would also want to add Z scores which would add two more columns for each explanatory variable. On the other hand, because the multi-year model has a much larger sample size (n=128) this analysis provides a more rigorous statistical analysis. Thus, when we report this model’s result, we provide the parameter values, +/- SEs, Z scores, and p-values so the reader can fully evaluate the significance of the model. 

Minor Changes:

>I have added line numbers to the text version of the ms as they were missing. 

Sorry for not including line numbers. I have added them in to my final draft. 

> Careful with the use of abbreviations such as HDOT. PLOS ONE guidelines state ‘Do not use non-standard abbreviations unless they appear at least three times in the text.’ 

I removed the abbreviation and spelt out the acronym. 

> in the CCT mentions throughout the text, remove the degree symbol before K (absolute temperature Kelvin is not used with the degree symbol)

 Thank you for making me aware of this. I have removed the degree symbol. 

> line 43. Remove comma after petrels: ‘petrels and puffins’ 

Done

> line 50. ‘conditions’ is repeated 

Thank you for catching this mistake. 

> line 60. Add reference regarding powerline collisions Travers et al. 2021 Avian Conservation Ecology 

Thank you for sharing this reference. I have added it.

> line 79. I disagree that a CCT >2700K is high. I would substitute by ‘recommended’ for example. Organism friendly. 

Thank you for this recommendation. I have changed the wording.

> line 292 and 293. Revise text. Two sentences are unconnected, ‘yet’ followed by ‘however’ is confusing. Thank you. Rephrased. 

> Table 1. Correct to ‘mean ratio (VMR).

 Not sure what is meant with this suggestion. The text states” variance to mean ratio”.

> Figure 3. Perhaps it would be appropriate to add a line plot with total of birds from the rescue centre.

We respectfully disagree because we have provided the yearly totals in S1 Table, and in Figure 2. 

> Figure 5. I find this particularly useful for the wind and moon integrated evaluation. The smaller circles represent 0 fallout? I would substitute all 0 fallout records with a different symbol, for example an ‘x’ or a ‘+’. As it is its not possible to discern surveys with no fallout or surveys with low fallout. I wonder if using two colors to represent the two light regimes would also be usefull in this graph. 

Thank you for this suggestion. We changed the zero values from an open circle to a small solid black dot, and edited the figure legend to explain the symbology. 

Comments from Reviewer 3

Comment 1: Page 4, last paragraph: “wedge-tailed shearwaters” should be capitalized.

Response: Thanks for catching that! Capitalized. 

Comment 2: Page 5, second line: USFWS should be spelled out the first time this acronym is used.

Response: Good point. Spelled out. 

Comment 3: Page 6, second paragraph, last sentence: The prediction that northeasterly winds would lead to higher fallout rates goes against the observation that more birds were found stranded during anomalous southerly winds in 1994, as stated on page 5, second paragraph. Please clarify. Also, it would be useful to add an explanation of why >8 mph winds was chosen as “strong”. For example, in other parts of the world, winds under 13 mph would be classified as low, so it would be good for the reader to have a better understanding of what typical wind speeds are for this particular region. Also, for consistency, it would be better to present all wind speeds as knots, rather than a combination of mph and knots.

Response: We have reworded our expectations by removing the 8 knot threshold and by explaining the link between NE winds and shearwater deposition along the SE shore of the island, downwind from two nesting colonies on offshore islets. Now, the text states: “Because WTSH rely on wind to take flight and may become disoriented in the absence of moonlight, we predicted higher fallout during windy nights of low moon illumination. In particular, due to the location of our study area, southwest from two breeding colonies, we anticipated that strong northeasterly winds would drive the fledging birds towards shore”. We provide an explanation below and can include similar wording in the ms, if you deem that it would help clarify our hypothesis.

When relating WTSH fallout to wind direction, we expect different patterns when we compare the counts of grounded birds within our small-scale study area versus the island-wide results. Work and Rameyer (1999) analyzed island-wide patterns in 1992 – 1994, using Sea Life Park (SLP) intake records. They reported widespread fallout in 1994, with WTSH distributed throughout the windward (east), leeward (south and west) and north shores of the island (see enclosed figure).

Yet, while WTSH fallout was widely distributed in 1994, when southerly winds deposited the birds throughout the entire island, large numbers of grounded birds were found within our study area, in the vicinity of the breeding sites on offshore islets (in the SE corner of the island). This result underscores the correlation we documented between SLP intake records and our road surveys. 

Thus, we hypothesize that strong winds lead to higher fallout. In particular, we feel like the tradewinds (NE winds) will deposit birds down-wind from the offshore islet colonies, leading to highly focused fallout in our study area, rather than widely-dispersed fallout throughout the island. 

Comment 4: Table 1: the p-values for moon and date were cut-off from the table in the produced document.

Response: We apologize if the table was not readable in this format. The journal submission guidelines stated not to reformat the table if it did not fit within the page dimensions, because it will be turned sideways and displayed in landscape orientation. 

Comment 5: While the Figure 4 caption is present in the text, the actual figure is missing from the document. 

Response: Apologies for erroneously leaving out Figure 4. We added it back in.

Additional Editorial Comments:

• Update formatting and file name - upload figures as individual files (Fig1.tif)

Done

• Add supporting information at the end of manuscript after refs 

Done

• 5. Please include a copy of Table A1 which you refer to in your text on page 11.

We removed Table A1, which originally referred to the table showing all tested model combos. Now, Table A1 refers to the fallout summary from the SLP Intake Records and the Road Surveys

• 6. Please upload a copy of Supporting Information Figure S1 and S2 which you refer to in your text on page 18 and 19 

We added these figures to the supplementary materials 

• Update the role of the funder

We updated the role of the funder 

• Make data publicly available

We provide a copy of the data used for the fallout models as S Table 2

• Make sure copyright info is all good for figure 1 map - added line in caption indicating that map was made in ArcGIS Pro and added ArcGIS Pro reference in references section. I made this map, it was no taken from another source.

---

## [Decision Letter · Decision Letter 1]

2 Feb 2022

PONE-D-21-31078R1Quantifying wedge-tailed shearwater (Ardenna pacifica) fallout after changes in highway lighting on Southeast Oʻahu, Hawaiʻi.PLOS ONE

Dear Dr. Urmston,

Thank you for submitting your manuscript to PLOS ONE. After careful consideration, we feel that it has merit but does not fully meet PLOS ONE’s publication criteria as it currently stands. Therefore, we invite you to submit a revised version of the manuscript that addresses the points raised during the review process.

We look forward to receiving your revised manuscript.

Kind regards,

Vitor Hugo Rodrigues Paiva, Ph.D.

Academic Editor

PLOS ONE

Journal Requirements:

Reviewers' comments:

Reviewer's Responses to Questions

**Comments to the Author**

1. If the authors have adequately addressed your comments raised in a previous round of review and you feel that this manuscript is now acceptable for publication, you may indicate that here to bypass the “Comments to the Author” section, enter your conflict of interest statement in the “Confidential to Editor” section, and submit your "Accept" recommendation.

Reviewer #1: All comments have been addressed

Reviewer #2: (No Response)

Reviewer #3: All comments have been addressed

2. Is the manuscript technically sound, and do the data support the conclusions?

Reviewer #1: Yes

Reviewer #2: Yes

Reviewer #3: Yes

3. Has the statistical analysis been performed appropriately and rigorously? 

Reviewer #1: Yes

Reviewer #2: Yes

Reviewer #3: Yes

4. Have the authors made all data underlying the findings in their manuscript fully available?

Reviewer #1: Yes

Reviewer #2: Yes

Reviewer #3: Yes

5. Is the manuscript presented in an intelligible fashion and written in standard English?

Reviewer #1: Yes

Reviewer #2: Yes

Reviewer #3: Yes

6. Review Comments to the Author

Reviewer #1: The revised manuscript is much improved with refined flow and focus on the change to LED lights. I only have a couple minor points:

Minor points:

Line 40 – remove unnecessary bracket.

Line 42 – USFWS should be spell out

Table 1 – spell out Wind Speed, as WS is not clear.

Line 275 – I am uncertain what do you mean by ‘larger degree angle’? Maybe try wind coming from land/offshore.

Line 289 – can you specify ‘a single new moon period’

Lines 406-408 – It could be also a result of other moon cycle variables (timing, location of moon in respect to earth) and environmental conditions (cloud cover) that were not taken into account in the model.

Figure 4 – I do not see solid black dots indicating absence of fallout.

Reviewer #2: First thank you to the authors for the detailed reply to the first revision comments. The current title is great, a significant improvement from the original submission. The last paragraph in the abstract also improved the overall idea of this ms: while it is good that the change in LED did not apparently alter fallout, this change could still affect other species and other locations differently. We are still in the early stages of such shift, and it will be critical to understand as broadly as possible how this LED change will affect ecosystems and species.

The discussion has been greatly improved, and now it reads like a study on the effects of lighting schemes changes, while environmental conditions were brought to a secondary placement.

I look forward to see the results of the geographical analysis in a future work!

Overall I only found some minor comments and a few notes, presented below. Otherwise good work, this is an important paper, an initial evaluation dealing with an emergent and widespread issue in light pollution.

Notes

Still unclear in the text regarding the state of collected birds: I understand from the authors that both live and dead birds were collected during the surveys, even if 99% of birds were dead. I maintain my recommendation to include this information in the ms, thus facilitate comparison with other studies and clarifying the work. For example: in methods explain that both live and dead birds were collected but since most were found dead (99%) only these were used for the analysis (unless all were used for the analysis, in which case please correct line 127 and other mentions of 'carcass' across the text); In the discussion, mention that even if this study only used dead birds, as you have found a good parallel between SLP intakes (island wide records of fallout, both live and dead) and this study, there is no evidence for a different effect of light change to the state of the birds, for example. These additions do not need to be lengthy, a short sentence will suffice.

Introduction: 4th paragraph (line 31-38) is a bit redundant, it could be shortened to be more concise and direct.

Minor corrections:

Abstract: 'due to exhaustion or collision' (use singular for collision).

Line 14 - missing word: 'especially in the absence of "moonlight"...'

Line 20 - 'that can affect seabirds' or better yet, 'that can affect fallout'?

line 64: 3000-4000 K LED lights (missing LED)

Line 64: I think (might be wrong) that is more accurate to say that CCT is the measure of how warm and cold a light appears, Kelvin is the unit.

Line 123: I would use full extent month names throughout the text (November 6 instead of Nov.). Or at least homogenize across the text.

Line 285: instead of 'throughout this study' something like 'within the time frame of this study'

Line 324: distinguish between these two factors

Line 337: I agree that evidence indicates stronger winds will increase number of fledgling. But wind affects groundings two ways a) increase the number of fledglings and b) push these inland (especially when southwestern winds are about correct?). I would add this last part. you can reference 10 and 11 refs [Rodríguez et al 2014 and Syposz et al 2018]

Reviewer #3: While the authors have addressed the previous round of comments, I have noticed some conflicting information in the introduction which I feel needs to be resolved as it impacts the logic of the main hypothesis. Line 34 states that light with high CCT (>2700K) is recommended for wildlife (note that a reference to support this statement is needed). Line 64 states that streetlights were changed from 2200K HPS lights to shielded 3000-4000 K LED lights (note that the word "LED" is missing in line 64 and should be added). Following the logic from the former statement (i.e., that light with >2700K is recommended for wildlife), the new LED light regime would be beneficial to shearwaters, however, the hypothesis states the opposite. Furthermore, the hypothesis is based on differences in wavelength rather than on CCT and K which are the light characteristics (related to temperature) discussed in detail in the introduction. Pulling all this together, it appears that the main hypothesis is based on the previous study of shearwaters which showed maximum light absorption by white LED lights emitting short wavelengths vs HPS lights which emit longer wavelengths, which presumably means that they are more attracted to white LED lights. As currently written, the introduction lacks a clear description of the relationship between the two light characteristics presented, namely: temperature and wavelength, and how these informed the predictions of the hypothesis.

Other minor edits:

Line 20: "seabird" should be plural

The sentence in lines 65 and 66 seems out of place as it talks about CCT while the previous sentence talks about K; the latter sentence needs to be tied somehow to the previous one.

7. PLOS authors have the option to publish the peer review history of their article (what does this mean?). If published, this will include your full peer review and any attached files.

Reviewer #1: No

Reviewer #2: **Yes: **Elizabeth Atchoi

Reviewer #3: No

---

## [Author Response · Author response to Decision Letter 1]

6 Mar 2022

Reviewer 1

Comment #1 Line 40 – remove unnecessary bracket. 

Response: Thank you for catching this mistake, I removed it. 

Comment #2 Line 42 – USFWS should be spell out. 

Response: Thank you for this suggestion, I have spelled it out. 

Comment #3 Table 1 – spell out Wind Speed, as WS is not clear. 

Response: Thank you for this suggestion. I have typed out wind speed in table #1. I left the abbreviation in table 2 as it is identified in the figure caption, and there is limited space in the column headings. 

Comment #4 Line 275 – I am uncertain what do you mean by ‘larger degree angle’? Maybe try wind coming from land/offshore. 

Response: Thank you for your comment. I agree that ‘larger degree angle’ is not very clear. I have changed it to say “wind blowing from the southwest” which I believe will be more intuitive to the readers. 

Comment #5 Line 289 – can you specify ‘a single new moon period’ 

Response: Changed to “a new moon period occurred in the middle of the fledging season, leading to a single peak in fallout”. This is opposed to the previous line describing a year with two new moon periods, early and late in the season. 

Comment #6 Lines 406-408 – It could be also a result of other moon cycle variables (timing, location of moon in respect to earth) and environmental conditions (cloud cover) that were not taken into account in the model.

Response: If the periodic moon cycle or other episodic environmental drivers (cloud cover and wind) were the drivers of fallout, we would expect that the 9-year (2002 – 2010) dataset of daily fallout records to yield a broad fallout distribution, caused by the overlapping of distinct annual peaks with different timing. Instead, we hypothesize that the consistent weekly fallout peak (Nov. 21 – 27) is caused by the timing of fledging, riven by the phenology of chick hatching and development. Nonetheless, I inserted more detail in this section: 

“When we replicated this analysis for WTSH, the quadratic model was not significant (R2=0.038, F2,5=1.140, p=0.39), suggesting that annual fallout did not follow the same pattern with the timing of the full moon. Although, other variables such as the timing of moon rise, cloud cover, and topography blocking the moon were not taken into account and may play a role in the moon’s influence on fallout.”

Comment #7 Figure 4 – I do not see solid black dots indicating absence of fallout. 

Response: Thank you for catching this. I changed it to “small solid dots”, they are grey not black. I have updated the figure caption. 

Reviewer 2

Comment #1

Still unclear in the text regarding the state of collected birds: I understand from the authors that both live and dead birds were collected during the surveys, even if 99% of birds were dead. I maintain my recommendation to include this information in the ms, thus facilitate comparison with other studies and clarifying the work. For example: in methods explain that both live and dead birds were collected but since most were found dead (99%) only these were used for the analysis (unless all were used for the analysis, in which case please correct line 127 and other mentions of 'carcass' across the text); In the discussion, mention that even if this study only used dead birds, as you have found a good parallel between SLP intakes (island wide records of fallout, both live and dead) and this study, there is no evidence for a different effect of light change to the state of the birds, for example. These additions do not need to be lengthy, a short sentence will suffice.

Response: Thank you for these suggestions. I have added some clarifying language to distinguish that our surveys focused on dead birds, and we only used dead birds in our analysis. 

“while visually searching for dead birds in each lane and along the shoulder. Since these surveys were conducted in the morning, likely a full 12 hours after fledging time, almost all the birds we observed were deceased. In 8 years of surveys, we observed 2 live birds, which were brought to Sea Life Park for rehabilitation and not counted in our analysis. All dead birds sighted”

“Our results are reassuring because they suggest that the shielded LED streetlights did not increase WTSH mortality due to fallout, as we hypothesized. Given the strong correlation between the dead birds observed in our road surveys and the live birds brought to SLP, there is no evidence suggesting that the shielded LED streetlights impacted the number of birds affected by fallout overall.”

Comment #2 

Introduction: 4th paragraph (line 31-38) is a bit redundant, it could be shortened to be more concise and direct. 

Response: Thank you for addressing this. I agree that this paragraph sounded misleading. I have rephrased this section to address the point that many newly implemented lights feature both shielding (good for birds) AND broad spectrum LEDs with high CCT values (potentially bad for birds). So the point I am trying to make is that we have a good change and a potentially bad change happening at the same time, and we don’t know how these changes coupled together are impacting birds. 

“Mitigation measures often target light directionality, whereby streetlights are shielded through the use of a “full-cutoff” design, which inhibits light emission above the horizontal plane of the fixture. This approach, when applied to HPS lights, reduced Newell’s Shearwater (Puffinus newelli) fallout on Kauai (Hawaiʻi) [16]. Although mitigation is being addressed through shielding, the common use of optimized LEDs with broad spectra and Correlated Color Temperature (CCT) greater than the maximum recommended value for wildlife (2200 K) may be a cause for concern [17]. While modern LED lights possess the flexibility to give off a range of low to high CTT, short-wavelength light with high CCT is a common choice because of its efficiency [19]. The effectiveness of light shielding coupled with the use of broad spectrum, high CCT LEDs is unknown.”

Comment #3

Abstract: 'due to exhaustion or collision' (use singular for collision). 

Response: Thank you for the suggestion, I have made this edit. 

Comment #4

Line 14 - missing word: 'especially in the absence of "moonlight"...' 

Response: Thank you for your thorough review and for catching this mistake! I have added in the word. 

Comment #5

Line 20 - 'that can affect seabirds' or better yet, 'that can affect fallout'? 

Response: Thanks for the suggestion, I have changed the phrasing to say “that can affect fallout”. 

Comment #6 

line 64: 3000-4000 K LED lights (missing LED) 

Response: Thank you, I have added in “LED”

Comment #7

Line 64: I think (might be wrong) that is more accurate to say that CCT is the measure of how warm and cold a light appears, Kelvin is the unit. 

Response: I think you are correct. I have rephrased this to say “where Kelvin (K) is a unit of measurement for CCT. Lower CCT indicates a warm yellow-orange appearance whereas higher CCT indicates cool blue light [18].”

Comment #8 

Line 123: I would use full extent month names throughout the text (November 6 instead of Nov.). Or at least homogenize across the text. 

Response: Thank you for pointing this out. I used full names in text and left abbreviations in tables only. 

Comment #9

Line 285: instead of 'throughout this study' something like 'within the time frame of this study' 

Response: Good suggestion, made change. 

Comment #10

Line 324: distinguish between these two factors 

Response: Thanks for the suggestion, I have made this edit. 

Comment #11 

Line 337: I agree that evidence indicates stronger winds will increase number of fledgling. But wind affects groundings two ways a) increase the number of fledglings and b) push these inland (especially when southwestern winds are about correct?). I would add this last part. you can reference 10 and 11 refs [Rodríguez et al 2014 and Syposz et al 2018] 

Response: I agree that this is likely the case, but this section is specifically referencing our multi-year model, which identified the Wind Speed * Moon interaction as the single most important variable, but did not include wind direction. Thus, we would like to avoid referring to wind speed and the advection of chicks inland. We are developing a second ms analyzing the spatial pattern of fallout in relation to wind speed and direction, and this ms will analyze and discuss those patterns. 

Reviewer 3

Comment #1 

Reviewer #3: While the authors have addressed the previous round of comments, I have noticed some conflicting information in the introduction which I feel needs to be resolved as it impacts the logic of the main hypothesis. Line 34 states that light with high CCT (>2700K) is recommended for wildlife (note that a reference to support this statement is needed). Line 64 states that streetlights were changed from 2200K HPS lights to shielded 3000-4000 K LED lights (note that the word "LED" is missing in line 64 and should be added). Following the logic from the former statement (i.e., that light with >2700K is recommended for wildlife), the new LED light regime would be beneficial to shearwaters, however, the hypothesis states the opposite. Furthermore, the hypothesis is based on differences in wavelength rather than on CCT and K which are the light characteristics (related to temperature) discussed in detail in the introduction. Pulling all this together, it appears that the main hypothesis is based on the previous study of shearwaters which showed maximum light absorption by white LED lights emitting short wavelengths vs HPS lights which emit longer wavelengths, which presumably means that they are more attracted to white LED lights. As currently written, the introduction lacks a clear description of the relationship between the two light characteristics presented, namely: temperature and wavelength, and how these informed the predictions of the hypothesis.

Response: Thank you for these comments. I agree that this section sounded misleading. I have rephrased this section to address the point that many newly implemented lights feature both shielding (good for birds) AND broad spectrum LEDs with high CCT values (potentially bad for birds). The point I am trying to make is that we have a good change and a potentially bad change happening at the same time, and we don’t know how these changes coupled together impacts birds. 

Also thank you for stating that a description of the relationship between wavelength and CCT is lacking, I have added this in the introduction: 

“Mitigation measures often target light directionality, whereby streetlights are shielded through the use of a “full-cutoff” design, which inhibits light emission above the horizontal plane of the fixture. This approach, when applied to HPS lights, reduced Newell’s Shearwater (Puffinus newelli) fallout on Kauai (Hawaiʻi) [16]. Although[JU1] [JU2] mitigation is being addressed through shielding, the common use of optimized LEDs with broad spectra and Correlated Color Temperature (CCT) greater than the maximum recommended value for wildlife (2200 K) may be a cause for concern [17]. While modern LED lights possess the flexibility to give off a range of low to high CTT, short-wavelength light with high CCT is a common choice because of its efficiency [19]. The effectiveness of light shielding coupled with the use of broad spectrum, high CCT LEDs is unknown.”

Additionally I corrected this line and provided a proper reference: 

Corrected this line - added reference: greater than the recommended value for wildlife (<2200 K) may be a cause for concern [17]. 

Other minor edits:

Comment #2

Line 20: "seabird" should be plural 

Response: I changed this to say “fallout” instead, as per Reviewer 2 recommendation 

Comment #3

The sentence in lines 65 and 66 seems out of place as it talks about CCT while the previous sentence talks about K; the latter sentence needs to be tied somehow to the previous one. 

Response: Thank you - rephrased 3000 – 4000 K LED lights, where Kelvin (K) is a unit of measurement for CCT. Lower CCT indicates a warm yellow-orange appearance whereas higher CCT indicates cool blue light [18].

---

## [Editor Report · Decision Letter 2]

9 Mar 2022

Quantifying wedge-tailed shearwater (Ardenna pacifica) fallout after changes in highway lighting on Southeast Oʻahu, Hawaiʻi.

PONE-D-21-31078R2

Dear Dr. Urmston,

We’re pleased to inform you that your manuscript has been judged scientifically suitable for publication and will be formally accepted for publication once it meets all outstanding technical requirements.

Kind regards,

Vitor Hugo Rodrigues Paiva, Ph.D.

Academic Editor

PLOS ONE
---

## [Editor Report · Acceptance letter]

15 Mar 2022

PONE-D-21-31078R2 

Quantifying wedge-tailed shearwater (*Ardenna pacifica*) fallout after changes in highway lighting on Southeast Oʻahu, Hawaiʻi. 

Dear Dr. Urmston:

I'm pleased to inform you that your manuscript has been deemed suitable for publication in PLOS ONE. Congratulations! Your manuscript is now with our production department. 

Kind regards, 

on behalf of

Dr. Vitor Hugo Rodrigues Paiva 

Academic Editor

PLOS ONE